# A lipid-binding protein mediates rhoptry discharge and invasion in *Plasmodium falciparum* and *Toxoplasma gondii* parasites

Catherine Suarez [1], Gaëlle Lentini [1,5], Raghavendran Ramaswamy[2,5], Marjorie Maynadier[1,5],
Eleonora Aquilini[1], Laurence Berry-Sterkers[1], Michael Cipriano [3], Allan L. Chen[4], Peter Bradley[4],
Boris Striepen[3], Martin J. Boulanger[2] & Maryse Lebrun[1]

Members of the Apicomplexa phylum, including *Plasmodium* and *Toxoplasma*, have two types of secretory organelles (micronemes and rhoptries) whose sequential release is essential for invasion and the intracellular lifestyle of these eukaryotes. During invasion, rhoptries inject an array of invasion and virulence factors into the cytoplasm of the host cell, but the molecular mechanism mediating rhoptry exocytosis is unknown. Here we identify a set of parasite specific proteins, termed rhoptry apical surface proteins (RASP) that cap the extremity of the rhoptry. Depletion of RASP2 results in loss of rhoptry secretion and completely blocks parasite invasion and therefore parasite proliferation in both *Toxoplasma* and *Plasmodium*. Recombinant RASP2 binds charged lipids and likely contributes to assembling the machinery that docks/primes the rhoptry to the plasma membrane prior to fusion. This study provides important mechanistic insight into a parasite specific exocytic pathway, essential for the establishment of infection.

[1] UMR 5235 CNRS, Université de Montpellier, 34095 Montpellier, France. [2] Department of Biochemistry and Microbiology, University of Victoria, Victoria, British Columbia V8W 3P6, Canada. [3] Department of Pathobiology, School of Veterinary Medicine, University of Pennsylvania, Philadelphia, PA 19104, USA. [4] Molecular Biology Institute, University of California, Los Angeles, Los Angeles, CA, USA. [5] These authors contributed equally: Gaëlle Lentini, Raghavendran Ramaswamy, Marjorie Maynadier. Correspondence and requests for materials should be addressed to M.L. (email: maryse.lebrun@umontpellier.fr)

Apicomplexan parasites are a serious threat to human health on a global scale. Of particular importance are the etiological agents of malaria (Plasmodium), cryptosporidiosis (Cryptosporidium), and toxoplasmosis (Toxoplasma). The control of these pathogens is problematic as no effective vaccines are available and drugs are either ineffective, poorly tolerated or rapidly becoming obsolete due to emerging resistance, the latter of which carries major implications for global efforts to eradicate malaria. Given the clear impact of these pathogens on human health and the challenges in effectively controlling them, novel treatments are needed.

Apicomplexa are obligatory intracellular parasites. They invade a wide variety of vertebrate and invertebrate hosts, often with species and cell specificities. Once in the intracellular environment, the parasite is surrounded by a parasitophorous vacuolar membrane (PVM) that is initially formed from the host plasma membrane during parasite invasion. After several rounds of replication, the parasite escapes the host cell and eventually reinvades a new cell. The ability of apicomplexan parasites to perform these different steps and to cause disease is critically dependent on the sequential secretion of proteins from phylum-specific organelles. The invasive stages of apicomplexans are indeed characterized by the presence of an apical complex composed of specialized cytoskeletal structures as well as two types of secretory organelles; the micronemes and rhoptries that sequentially release their contents during invasion[1]. Micronemes are small rod-shaped organelles that cluster at the apical pole of the parasite. Their exocytosis is initiated during the intracellular cycle of the parasite by sensing the loss of host-cell integrity, a process that involves intrinsic and extrinsic signalling events and parasite effectors recently described in great detail for Toxoplasma (reviewed in Bullen et al.[2]). Microneme release is associated with the translocation of microneme proteins (MICs) at the parasite plasma membrane where MICs contribute to motility, invasion and egress (reviewed in Frenal et al.[3]). The release of rhoptries immediately follows microneme exocytosis, suggesting a tightly coordinated signalling pathway between the two events, although the details remain elusive.

Rhoptries are almost ubiquitous throughout the phylum Apicomplexa. They are club-shaped organelles, with three sub-compartments; the bulb, the neck and the intersection between the bulb and neck[4,5]. They are located at the anterior pole of the parasites, and their number varies between parasites[4]; P. falciparum merozoites contain two whereas T. gondii possesses 8–12 rhoptries clustered together. In T. gondii, the apical complex also harbours a microtubule-rich spiral structure (conoid) and only one or two rhoptries can access the internal part of the conoid to apically dock their neck for secretion[6]. Rhoptry secretion is induced upon contact with the host cell and results in the injection of the rhoptry contents into the host cytoplasm. Once secreted, rhoptry proteins take part in multiple and distinct processes throughout the parasite life cycle. In both, Plasmodium and Toxoplasma, rhoptry neck proteins assemble the moving junction (MJ)—the mechanistic core of the apicomplexan invasion machine[3,7]. In Plasmodium, rhoptry bulb proteins have been shown to contribute to invasion[8], facilitate parasite survival in its intracellular environment[9] and make the red blood cell more permeable to absorb nutrients from the blood stream[10,11]. In Toxoplasma, a large family of rhoptry bulb kinases disable multiple host cell innate restriction mechanisms[12]. In the past two decades, our knowledge of the effectors delivered has expanded considerably and positioned the rhoptries at the heart of the pathogenesis of Apicomplexa. However, the signalling pathways for rhoptry discharge, the molecular mechanisms governing rhoptry docking and fusion to the extreme apical end of the parasite (exocytosis), and the subsequent export of rhoptry contents across the host cell membrane, remain enigmatic. This is in stark contrast to the abundant literature describing the secretion mechanism of the microneme organelles[2,3]. Rhoptry secretion has been difficult to study since this event is essential for the parasite lifecycle and no triggers or inducers have yet been identified to facilitate biochemical approaches. Currently, rhoptry secretion can only be observed during the process of invasion, suggesting that this relies on direct recognition between the surface molecules of the parasite apex and receptor molecules on the host cell membrane. In support of this hypothesis, the microneme protein EBA175, released at the surface of the merozoites of P. falciparum, binds Glycophorin A at the red blood cell surface. This event is associated with rhoptry secretion[13]. In Toxoplasma, the microneme protein MIC8 is similarly involved in rhoptry secretion, but the mechanism (e.g., binding to a host receptor) is unknown[14]. Thus, rhoptry exocytosis is likely connected to microneme secretion and parasite-host adhesion. It is therefore challenging to perform bulk genetic screens to uncover those multi-step complex mechanisms via mutants considering the essentiality of these events.

In the present study, we use computational analyses of rhoptry gene expression and a yeast genetic screen to identify a set of proteins that localise to the rhoptry surface and cap the apex of the organelle. Based on this newly described localisation, we refer to these proteins as RASPs for Rhoptry Apical Surface Proteins. Subsequent genetic and cell biological experiments in T. gondii and P. falciparum demonstrate that RASP2 plays a key role in rhoptry exocytosis. Conditional depletion of RASP2 ablates rhoptry secretion, blocks host cell invasion and renders the parasites unable to propagate, highlighting the importance of this conserved mechanism in Toxoplasma and Plasmodium. Furthermore, biochemical and genetic studies, supported by molecular modelling, reveal that RASP2 binds specifically to PA and $PI(4,5)P_2$ through a calcium lipid-binding-like domain (C2) and a Pleckstrin Homology-like domain (PH) that we demonstrate to be crucial for rhoptry secretion. Our findings support a role for RASP2 in docking/priming the rhoptries to the parasite plasma membrane.

## Results

**RASPs display an unprecedented localisation on *Toxoplasma* rhoptries.** The expression of rhoptry proteins coincides with the formation of the rhoptry organelle during the final phase of parasite replication[15]. We exploited this in a bioinformatics screen in T. gondii to identify new rhoptry-related proteins based on their transcriptional profile over the replication cycle. We initially identified TGGT1_235130, a protein previously reported to be associated with centrosomes[16] (Fig. 1a). Here we show that TGGT1_235130 partially co-localises with ARO (Fig. 1b and Supplementary Fig. 1a), a protein associated with the cytoplasmic face of the rhoptry neck and bulb[17]. Remarkably, TGGT1_235130 exhibits an unusual localisation on the rhoptry, capping the extremity of the neck of the organelles (see inset in Fig. 1b and super-resolution microscopy 3D reconstruction in Fig. 1c and Supplementary Movie 1). A TGGT1_235130 paralogue, TGGT1_315160, displays similar localisation (Fig. 1b) but is expressed slightly later in replication (Fig. 1a). We found that the early TGGT1_235130 associates with pre-rhoptry compartments (stained by pro-ROP4) as well as mature rhoptries, whereas the late TGGT1_315160 only associates with fully mature organelles (Fig. 1b, lower panels, Supplementary Fig. 1a and b). Both proteins are predicted to be soluble (ToxoDB.org), but conflicting predictions for the presence of a signal sequence were obtained. We then performed differential permeabilization and protease accessibility assays and demonstrated that both proteins localise

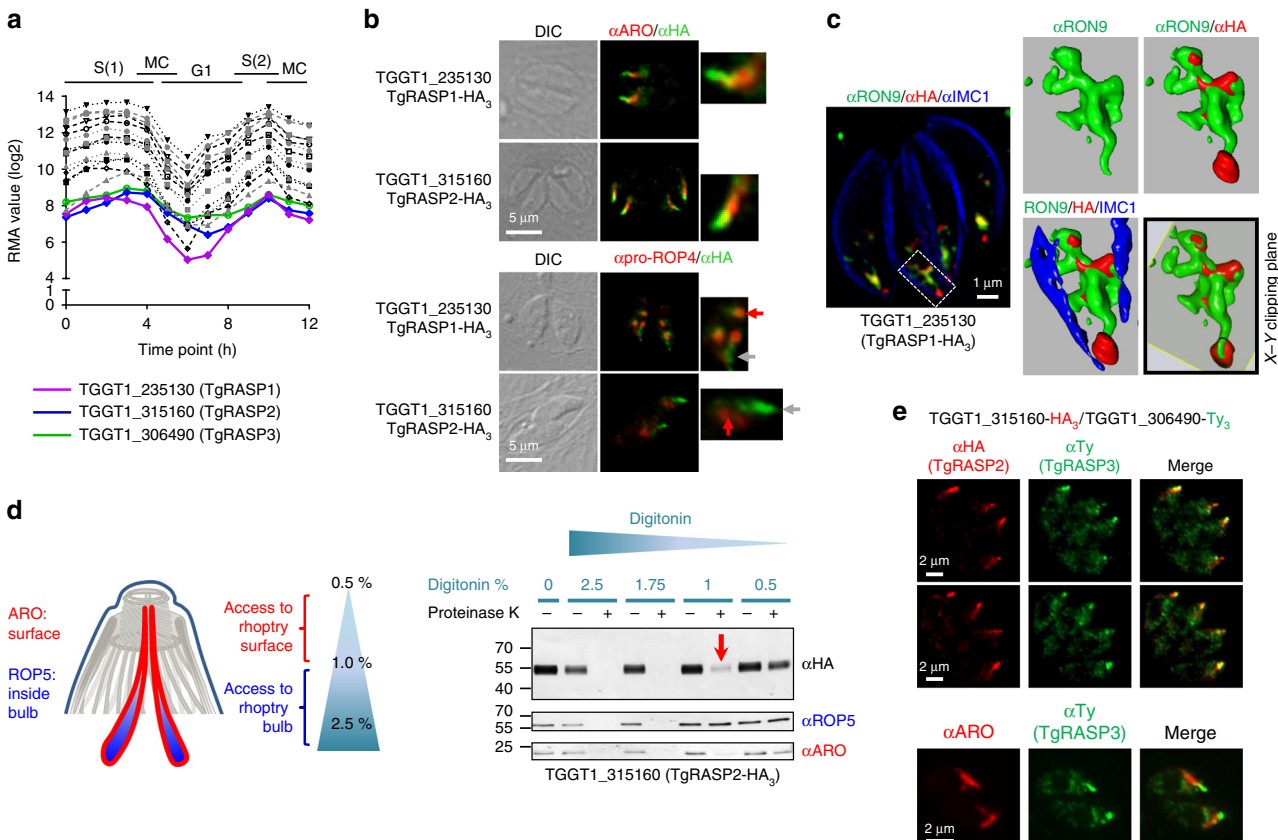

**Fig. 1** A subset of novel rhoptry surface proteins define a new apical rhoptry sub-compartment. **a** Comparison of the expression pattern of known rhoptry proteins during the tachyzoite cell cycle in combination with the expression profile of TGGT1_235130 (TgRASP1), TGGT1_315160 (TgRASP2) and TGGT149_306490 (TgRASP3). Data obtained from ToxoDB[15]. **b** Immunofluorescence assay (IFA) of HA-tagged TGGT1_235130 (TgRASP1) and TGGT1_315160 (TgRASP2) tachyzoites. Insets/higher magnifications: localisation of the proteins overhanging the rhoptry neck (upper panels) and the presence of TgRASP1 in pro-rhoptries (red arrows). Grey arrows indicate mature rhoptries. **c** Super-resolution microscopy on TgRASP1-HA₃ tachyzoites. Inset: 3D reconstruction of the apical region shows how the rhoptry neck is capped (see X-Y clipping plane). **d** Schematic representation of the differential permeabilization assay and localisation of ROP5 and ARO. Immunoblots of TgRASP2-HA₃, ROP5, and ARO show that, TgRASP2 is accessible to proteinase K digestion after permeabilization with 1% digitonin (arrow) similarly to the rhoptry surface protein ARO. **e** Two upper panels: IFAs of TgRASP2-HA₃ and TgRASP3-Ty₃ parasites using anti-HA and anti-Ty antibodies reveal co-localisation of the two proteins. Lower panel: co-localisation of ARO and TgRASP3-Ty₃. TgRASP3 is present at the extremity of the rhoptry, however additional staining in the cytosol is also observed for this protein. Source data are provided as a Source Data file

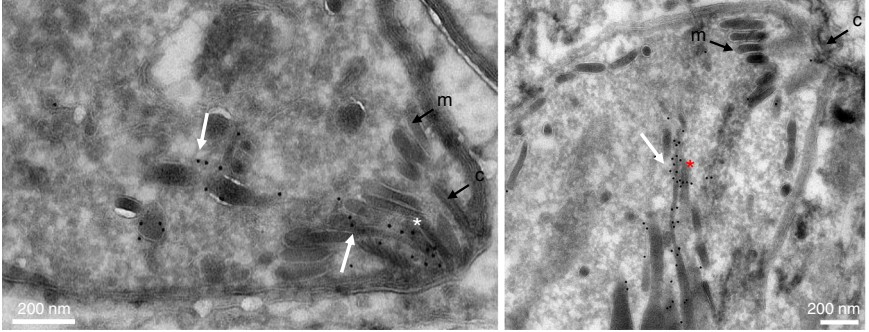

**Fig. 2** TgRASP2 accumulates at the extremity of rhoptries. Immuno-electron microscopy of TgRASP2-HA₃ tachyzoites using an anti-HA antibody. The white arrows point to the gold particles associated with the rhoptries. The white asterisk indicates the rhoptry neck inside the conoid and the red asterisk the accumulation of gold particles at the extremity of a rhoptry. m: micronemes, c: conoid

to the surface of the rhoptries (Fig. 1d and Supplementary Fig. 1c), and thus do not enter the secretory pathway. Immuno-electron microscopy further confirmed the localisation of TGGT1_315160 to the rhoptry surface (Fig. 2, white arrowhead, Supplementary Fig. 1d), including around the neck of rhoptries

already engaged inside the conoid (Fig. 2, white asterisk). TGGT1_235130 and TGGT1_315160 contain several predicted palmitoylation sites; mutagenesis of those in TGGT1_315160 did not change protein localisation suggesting a mechanism for rhoptry association other than palmitoylation (Supplementary

Fig. 1e and f). We also showed that TGGT1_235130 and TGGT1_315160 co-localise and interact by co-immunoprecipitation studies (Supplementary Fig. 2a and b). A third member (TGGT1_306490) of this complex was identified in a yeast two-hybrid screen using TGGT1_315160 as the bait against a *Toxoplasma* cDNA library (Supplementary Table 1), and its interaction confirmed by co-immunoprecipitation (Supplementary Fig. 2c). Notably, TGGT1_306490 is also present at the extremity of the rhoptry neck (Fig. 1e).

Together, these results define a subset of new rhoptry proteins that predominantly accumulate at the very apical tip of the organelle, which, based on this newly described localisation, led us to designate them as TgRASP1 (TGGT1_235130), TgRASP2 (TGGT1_315160) and TgRASP3 (TGGT1_306490) for *T. gondii* rhoptry apical surface proteins.

**The RASP ortholog in *P. falciparum* is expressed at late schizont stage**. While TgRASP3 is specific to coccidians, orthologues of TgRASP1 and 2 are conserved across the phylum (Supplementary Fig. 3a). *P. falciparum* encodes one RASP related protein, PF3D7_0210600 with high sequence similarity to TgRASP1 and TgRASP2. Matching our observations for TgRASP2 in *T. gondii*, c-terminal endogenous tagging of PF3D7_0210600 shows that this protein is expressed later than the rhoptry neck marker PfRON4 and the rhoptry bulb marker PfRhopH2 (Fig. 3a, Supplementary Fig. 4a–d) as predicted by its transcriptional profile (Fig. 3b). Additionally, it co-localised with PfRON4 but only partially with PfRhopH2 (Fig. 3c) indicating that PF3D7_0210600 is associated with the apical part of rhoptries in *Plasmodium* merozoites. Based on the functional similarity with TgRASP2 (see below), we refer to this protein hereafter as PfRASP2. Following invasion, PfRASP2 staining remains as a dot within the apical end of parasite (Fig. 3d) while PfRhopH2 and PfRON4 are secreted and relocalised to the parasitophorous vacuole membrane and MJ respectively[18], suggesting a function different from previously reported rhoptry proteins.

***Toxoplasma* RASP2 is essential for tachyzoite invasion**. To unravel the function of *Toxoplasma* RASP proteins, we next generated *T. gondii* mutants. We removed *tgrasp1* by double cross-over and generated a conditional knockdown of *tgrasp3* using the auxin-inducible degron system[19] (Supplementary Figs. 5 and 6). No defects in growth, invasion or replication were observed for *tgrasp1* deletion and *tgrasp3* knockdown (Supplementary Figs. 5 and 6), consistent with the fitness score detected in a genome-wide CRISPR/Cas9 screen[20]. In contrast, we were unable to remove *tgrasp2*. We therefore conditionally depleted TgRASP2 using the tetracycline-inducible repressing system[21] (Fig. 4a–c, Supplementary Fig. 7a and b) and showed that KD-TgRASP2 parasites were unable to form lysis plaques on fibroblast monolayers in the presence of anydrotetracycline (ATc) (Fig. 4d), reflecting the inability of the mutant to complete the lytic cycle in the absence of TgRASP2. Importantly, this phenotype was fully rescued by adding a complementing copy of *tgrasp2* (cKD-TgRASP2 parasites) (Fig. 4d and Supplementary Fig. 7b and c). In TgRASP2-depleted parasites, intracellular replication, parasite egress and motility occurred normally (Fig. 4e–g) but invasion was reduced by 98.8% (Fig. 4h). Finally, we checked the localisation and expression of RASP1 and RASP2 in the mutants and showed that their localisation is not dependent on each other (Supplementary Fig. 5g and h and Supplementary Fig. 7d and e).

In conclusion, while depletion of TgRASP1 and TgRASP3 proteins do not impact on the ability of the mutants to growth in

fibroblasts, TgRASP2 plays a critical role for invasion in *Toxoplasma*.

***P. falciparum* RASP2 is essential for merozoite invasion**. To generate a *P. falciparum* RASP2 mutant (iKO-PfRASP2), we used the rapamycin-inducible dimerizable Cre recombinase (DiCre) system[22] to conditionally excise the *pfrasp2* gene. For this, we used Cas9-mediated genome editing to flox and tag *pfrasp2* simultaneously (Fig. 5a, Supplementary Fig. 8a and b). Successful excision of the *pfrasp2* gene was induced prior to PfRASP2 expression by adding rapamycin at the ring stage followed by removal 6 h later (Fig. 5b). Loss of the protein in late schizonts was confirmed by Western blot and IFA (Fig. 5c, d). Monitoring parasitaemia, we next showed that loss of PfRASP2 effectively cured infected cultures (Fig. 5e), highlighting the critical role of PfRASP2 for parasite survival. Consistent with the onset of PfRASP2 expression in late schizonts (Fig. 3b), the intracellular development of the PfRASP2-depleted parasites was not affected during the first cycle (Fig. 5e) and merozoites egressed normally (Fig. 5f, Supplementary Movies 2 and 3). However, these merozoites were unable to reinvade new erythrocytes (Fig. 5g).

**RASP2 is required for rhoptry secretion**. Host cell invasion by Apicomplexa is critically dependent on the sequential secretion of secretory organelles. Therefore, to gain further insight into the invasion phenotype of the KD-TgRASP2 and iKO-PfRASP2 parasites, we first tested the ability of the mutants to secrete the contents of micronemes, a crucial step for attachment, motility and subsequent rhoptry secretion. To accomplish this, we analysed the release of the micronemal protein AMA1 in the culture medium of *Toxoplasma* (Fig. 6a) and *P. falciparum* (Fig. 6b), and the PfAMA1 translocation from the micronemes to the plasma membrane of fully mature daughter merozoites during egress (Fig. 6c). In both mutants, microneme secretion was unaffected, suggesting that the rhoptry secretion defect observed is independent of the ability of the mutant to secrete proteins from micronemes. In contrast, a complete inhibition of rhoptry discharge in the host cell was observed in the absence of TgRASP2 or PfRASP2, a feature that correlates with the invasion defect. *Toxoplasma* RASP2 mutants were unable to export the rhoptry protein TgROP1 into the host cell (Fig. 6d) or to induce phosphorylation of the human STAT6 protein upon export of the rhoptry kinase ROP16 (Fig. 6e), a phenotype fully restored in the complemented strain. Similarly, we observed a complete inhibition of *P. falciparum* rhoptry bulb protein PfRAP2 export into the erythrocyte for PfRASP2-depleted parasites (Fig. 6f). Importantly, rhoptry morphology and positioning appeared unaffected in the absence of TgRASP2 (Fig. 4c, + ATc) or PfRASP2 (Fig. 5c, + rapa) by IFA, which we confirmed by electron microscopy on KD-TgRASP2 parasites (Fig. 6g). Thus, the rhoptry secretion defect was not caused by abnormal rhoptry biogenesis or defective tethering to the parasite apex. Altogether, these results suggest that these mutant parasites were impaired in rhoptry exocytosis.

**Lipid binding domains of RASP2 are involved in rhoptry secretion**. Apicomplexan RASP proteins lack immediately recognizable domains based on primary sequence analysis. However, secondary structure based predictions[23] revealed two putative lipid-binding domains, namely C2 ($Ca^{2+}$-dependent lipid-binding) and PH (Pleckstrin homology domain) domains (Fig. 7a and Supplementary Note 1) that are conserved across apicomplexan RASP2 proteins (Supplementary Fig. 3a). Homology modelling of TgRASP2 also predicted the presence of C2 and PH domains (Supplementary Fig. 3b), which are known

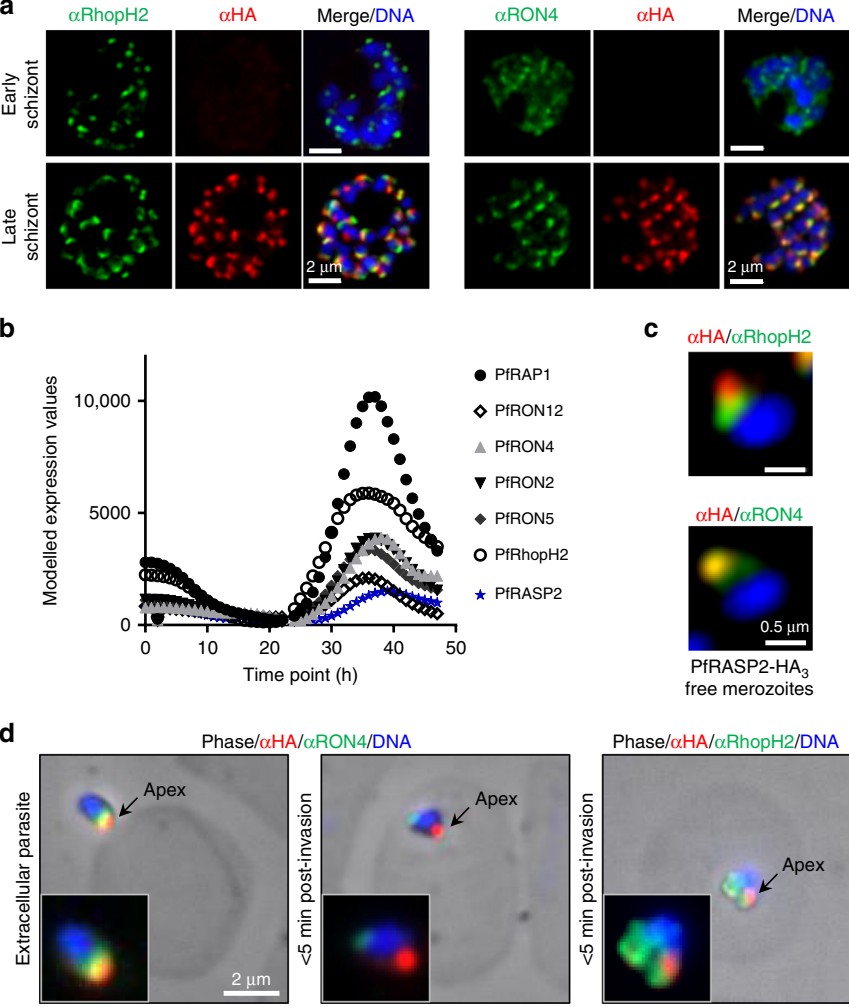

**Fig. 3** *Plasmodium falciparum* RASP2 expression and localisation in erythrocytic stage. **a** IFAs on PfRASP2-HA₃ schizonts showing PfRASP2-HA₃ with rhoptry markers. RON4, rhoptry neck, RhopH2, rhoptry bulb. **b** Comparison of the expression patterns of known rhoptry proteins during the *P. falciparum* erythrocytic cycle in combination with the expression profile of PF3D7_0210600 (PfRASP2). Real-time transcription data of the *P. falciparum* 3D7 strain obtained by biosynthetic pyrimidine labeling (PlasmoDB.org). **c** IFAs of PfRASP2 in free merozoites. **d** IFAs of invading merozoites. Left, extracellular parasite attached to red blood cell. Middle, at early point of invasion, the RON4 marker detects the moving junction at the posterior end of the intracellular merozoite, while RASP2 remains apical. Right, export of the bulb marker RhopH2 upon invasion. **c** Pictures of individual merozoites where acquired with a confocal microscope with Airyscan

mediators of phospholipid interactions. Of particular note were several regions modelled as surface loops that harboured hydrophobic and basic residues, which in canonical C2 and PH domains are responsible for phospholipid binding (Supplementary Fig. 9a–c). In contrast, the complement of aspartate residues that mediate calcium binding in canonical C2 domains is absent in TgRASP2 and therefore TgRASP2 is likely to function as a $Ca^{2+}$ independent interaction module (Supplementary Fig. 9d, Supplementary Note 2). In addition to the predicted C2 and PH domains, a conserved stretch of basic residues (designated as the polybasic region, PBR) may also support lipid binding (Fig. 7a and Supplementary Fig. 3a). We next sought to experimentally establish the lipid and calcium binding capacity of TgRASP2. To achieve this, we used a recombinant form of TgRASP2 (rTgRASP2con2 containing C2 and PH domains- Fig. 7a) in calcium binding and lipid-blot assays. Consistent with the lack of key aspartate residues, thermostability measurements using differential scanning fluorimetry (DSF) showed no change with rTgRASP2con2 following addition of different concentrations of either $Ca^{2+}$ or

EDTA (Supplementary Fig. 9e) indicating that $Ca^{2+}$ was not able to bind and stabilize the protein. These results were confirmed by isothermal titration calorimetry (ITC) measurements that detected no interaction between rTgRASP2con2 and $Ca^{2+}$ (Supplementary Fig. 9f). Collectively, our in silico and in vitro analysis is consistent with a $Ca^{2+}$-independent function of TgRASP2. Lipid blots, however, showed that rTgRASP2con2 bound several species of phosphoinositides (PIPs) and phosphatidic acid (PA) (Supplementary Fig. 9g). To validate the PIP-strips experiments in the context of a membrane bilayer, we performed liposome binding assays and showed that rTgRASP2con2 binds PA and phosphatidylinositol 4,5-bisphosphate (PIP2) but not phosphatidylinositol 4-phosphate (PI4P) (Fig. 7b, c). We next mutated the basic residues on the surface loops of the putative C2 and PH domains to establish the role of these residues in the mechanism of binding (Supplementary Fig. 3b). The mutated version of TgRASP2 (TgRASP2con2^MUT) was not able to bind to either PA or PIP2 in the liposome binding assay demonstrating the importance of these basic residues for the interaction of TgRASP2 with lipids (Fig. 7b, c).

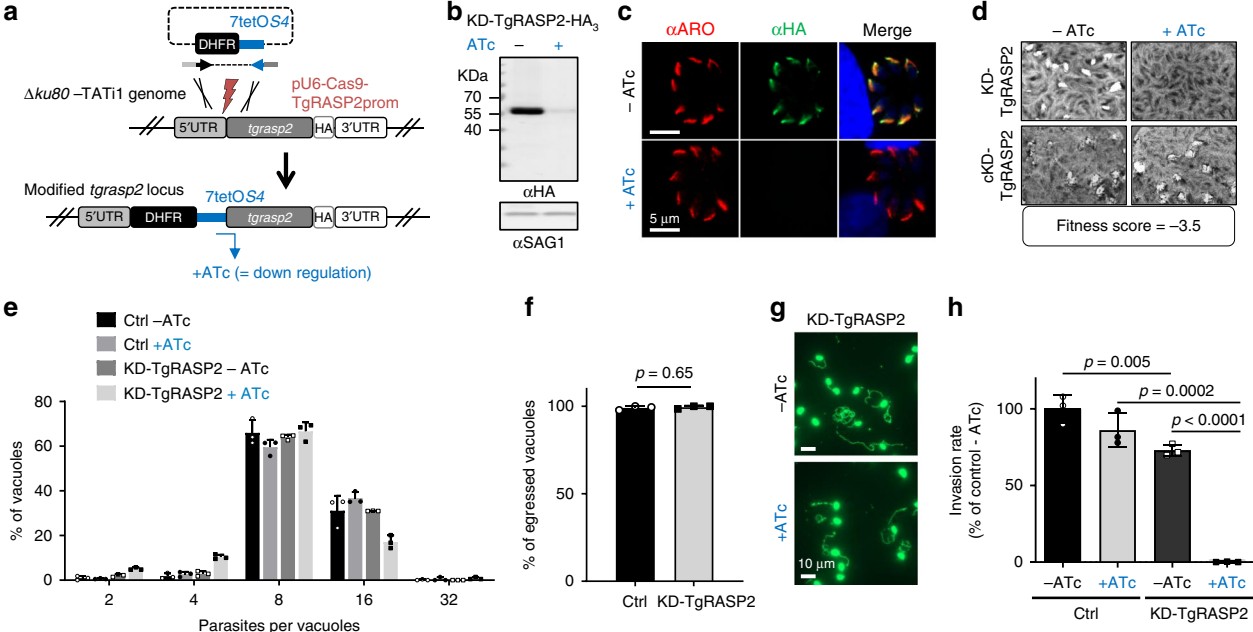

**Fig. 4** *Toxoplasma* RASP2 depletion impairs parasite invasion. **a** Strategy for conditional depletion of TgRASP2 using the Tet-OFF system. **b** Immunoblot of KD-TgRASP2-HA$_3$ (±ATc). SAG1, loading control. **c** IFA of KD-TgRASP2-HA$_3$ shows depletion of RASP2 after 2 days ATc treatment. **d** Plaque assays of KD-TgRASP2 and complemented cKD-TgRASP2 parasites (±ATc) shows that the strong phenotype induced by TgRASP2 depletion (no plaques) can be rescued by a complementing copy of the gene. CRISPR fitness score was derived from Ref. [20]. **e** Intracellular replication of TATi_TgRASP2-HA$_3$ (Ctrl) and KD-TgRASP2-HA$_3$ ± 48 h ATc. The percentage of vacuoles containing 2, 4, 8, 16 or 32 parasites was determined on 200 vacuoles (n = 3). **f** Percentage of egressed vacuoles in TATi_TgRASP2-HA$_3$ (Ctrl) and KD-TgRASP2-HA$_3$+ATc. Parasite egress was induced 30 h post-invasion by addition of A23187 (3 µM) for 8 min before samples were fixed and processed for IFA (anti-GRA3 antibodies). Egress events (GRA3 in the PV) were quantified on 30 vacuoles. **g** Gliding assays were performed with KD-TgRASP2 ± ATc 48 h. Trails were revealed by IFA using anti-SAG1 antibodies. **h** 5 min-invasion assay of TATi_TgRASP2-HA$_3$ (Ctrl) and KD-TgRASP2-HA$_3$ parasites (±ATc) shows a strong invasion defect when TgRASP2 is affected. (d, e, f, h) Values represent means ± SD, n = 3, from a representative experiment out of 2 independent assays. Source data are provided as a Source Data file

Finally, to define which part of TgRASP2 is important for rhoptry secretion, we complemented the KD-TgRASP2 with truncated versions of TgRASP2. We first showed that the deletion of C2 results in a complete mislocalisation of TgRASP2 to the cytoplasm (Supplementary Fig. 10, Supplementary Note 3). Neither the individual deletion of the PH or PBR domains nor the combination of both abrogated rhoptry trafficking or affected the ability of tachyzoites to grow in HFF cells. However, removal of the three basic residues within the putative lipid binding loop 3 of the C2 domain in the ΔPH mutant (L3*_ΔPH in Fig. 7d) completely abrogated the function of TgRASP2, despite the protein being localised to the rhoptries (Supplementary Fig. 10). In contrast, no rhoptry secretion defect was observed solely by mutating the loop 3 (L3* in Fig. 7d).

Collectively, these results showed that TgRASP2 specifically binds PA and PIP2 and revealed an essential cooperative role of the PH domain and the loop 3 in the C2 domain for rhoptry secretion and parasite survival in fibroblasts.

## Discussion

In stark contrast to the abundant literature describing mechanistic details of secretion from the microneme organelles[2], the secretion machinery that delivers rhoptry effectors remains poorly understood. Here we fill a core knowledge gap by describing an Apicomplexa specific protein that caps the extremity of rhoptries and plays a role in rhoptry exocytosis in both *Toxoplasma* and *Plasmodium*. We show that RASP2 contains C2 and PH-like domains that contribute to PA and PIP2 binding and cooperate for rhoptry secretion in a Ca$^{2+}$-independent manner. Interestingly, the presence of lipid binding domains such as C2

and PH domains are a hallmark of proteins that act as regulators of calcium-dependent dense core vesicle exocytosis[24]. By binding to calcium and lipids, these proteins trigger the docking and/or priming of vesicles for fusion. Both PA and PIP2 are lipids that accumulate at exocytic sites and participate in membrane fusion events during vesicle exocytosis[25,26]. In *Toxoplasma* prior to egress, PA is generated at the apical end of the parasite plasma membrane and recognised by APH, a protein present on the microneme surface that initiates membrane fusion for microneme exocytosis[27]. Similar to what was established for APH, we propose that, following microneme secretion, RASP2 binds to PA and or PIP2 at the apex of the parasite plasma membrane (Fig. 8). This interaction would facilitate the close apposition of the rhoptry to the parasite plasma membrane, leading to subsequent recruitment of a hypothetical membrane fusion machinery, e.g., SNARE proteins. Our model places RASP2 at the first exocytic step of the complex mechanism of rhoptry secretion (Fig. 8, panels A and B), but leaves unknown how the content of the organelle is injected into the host (panel C), a fully enigmatic process. Indeed, exocytosis of the rhoptry and export of its content are intimately connected both physically and temporally. An opening at the very apex of *Toxoplasma* tachyzoites is observed during invasion and likely corresponds to the opening formed by the fusion event of rhoptry and parasite plasma membranes[6]. A puzzling question remains how the rhoptry content is injected inside the host cell across the host plasma membrane. Freeze fracture analysis of invasion events suggests an early opening pore of 40 nm inside the host[28]. This hypothesis is supported by electrophysiology studies of host cells during *Toxoplasma* invasion, that revealed a transient spike of conductance a few seconds before the parasite starts to enter the host[29]. Whether the

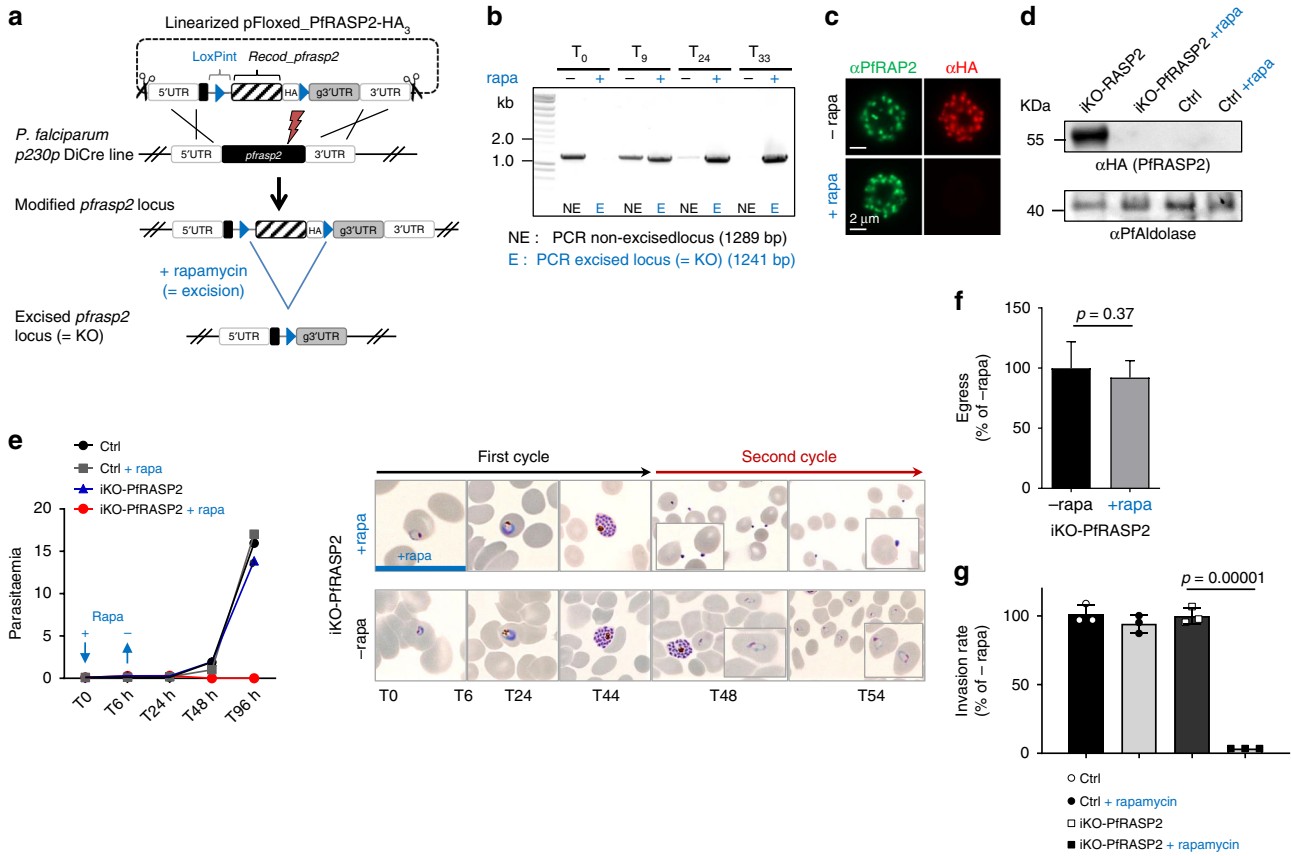

**Fig. 5** *P. falciparum* RASP2 is necessary for invasion. **a** Strategy for conditional excision of the *pfrasp2* gene using the DiCre system. **b** PCRs showing efficient excision of the *pfrasp2* locus upon addition of rapamycin with primers 3181/2911 (non-excised), 3181/2912 (excised). **c** IFA on iKO-PfRASP2-HA₃ schizonts ± rapamycin. **d** Immunoblot on schizont lysates from the *p230p* DiCre (parental line) and iKO-PfRASP2 mutant line ± rapamycin. PfAldolase, loading control. **e** Left: Growth curves (parasitaemias) of *p230p* DiCre (Ctrl) and iKO-PfRASP2 mutant ± rapamycin shows that PfRASP2-depleted parasites are unable to proliferate. Right: Giemsa stainings illustrating the development and reinvasion of iKO-PfRASP2 merozoites (48 h) in the absence of rapamycin and their accumulation at the surface of RBCs in the presence of rapamycin treatment. **f** Graphical summary of egress data of control (-rapa) and rapamycin-treated iKO-PfRASP2 schizonts following removal of C2. Data collected from 8 control movies (-rapa) and 11 movies in the presence of rapamycin. Number of egress events normalized as a percentage of control (-rapa) considered as 100% egress here. **g** Invasion rate of Ctrl and iKO-PfRASP2 parasites treated ± rapamycin. Synchronized cultures were treated for 6 h at ring stage with rapamycin or DMSO and were quantified by flow cytometry (100,000 RBCs were analysed for each time point) over 2 cycles (see methods). (e, g) Values represent means ± SD (*n* = 3) from a representative experiment out of two independent assays. Source data are provided as a Source Data file

formation of this opening pore is linked to export of rhoptry content into the host and if there is a continuity between this content and the cytoplasm of the host remains to be investigated.

While RASP2 sequences do not contain known SNARE binding motifs it is possible that such binding motifs exist but are too divergent to be detected by prediction software, or that RASP2 domains recognize unconventional apicomplexan SNARE proteins. Indeed, SNAREs are conserved from human to Apicomplexa[30], but none have yet been associated with organelle exocytosis in Apicomplexa. It is also conceivable that other proteins facilitate membrane fusion in the absence of SNAREs. A recent study identified TgFER2, a homolog of $Ca^{2+}$-sensing membrane fusion proteins of the ferlin family, that controls rhoptry secretion in *Toxoplasma*[31]. In mammals, otoferlin localises to synaptic vesicles and the plasma membrane in cochlear hair cell and might control the release of neurotransmitters independently of neuronal SNAREs[32]. The architecture of RASP2 is conserved throughout the phylum Apicomplexa. Moreover, its domains are highly divergent from its mammalian counterparts, suggesting that part of the priming and fusion machinery of rhoptries with the parasite plasma membrane is atypical. Since the discharge of rhoptry contents is essential for invasion,

nutrient import and the control of host immunity, this study identifies a new mechanistic framework for the development of therapeutics targeting rhoptry exocytosis.

## Methods

**Parasite culture**. Tachyzoites of the *T. gondii* RH strain deleted for the *ku80* gene (*Δku80*)[33] and expressing the transactivator TATi1[34] were used throughout the study. For auxin-induced degradation, a TIR1-expressing line was used. All tachyzoites were passaged in human foreskin fibroblasts (HFFs) (American Type Culture Collection-CRL 1634) or Vero cells (American Type culture Collection CCL 81) grown in Dulbecco's modified essential medium (Gibco-BRL), supplemented with 5% foetal calf serum (FCS) and 2 mM glutamine.

*P. falciparum* 3D7 *p230p* DiCre parasites[35] were grown in human erythrocytes obtained by donations from anonymized individuals from the French Blood bank (Establissement Français du Sang, Pyrénées Méditerranée, France) in RPMI 1640 medium (Gibco), supplemented with gentamycin at 20 µg/ml and 10% human serum. Parasite cultures were kept synchronised by 5% (w/v) sorbitol. The cultures were kept at 37 °C under a controlled tri-gas atmosphere (5% $CO_2$, 5% $O_2$ and 90% $N_2$).

Late stage schizonts were purified from highly synchronous cultures on cushions of 70% (v/v) isotonic Percoll adjusted to isotonicity and used for immunoblots, rhoptry and microneme secretion assays, invasion assays, immunofluorescence assays and time-lapse microscopy.

To induce DiCre recombinase mediated excision of the floxed *pfrasp2* locus, early ring stage parasites were treated with 10 nM rapamycin for 6–8 h, before

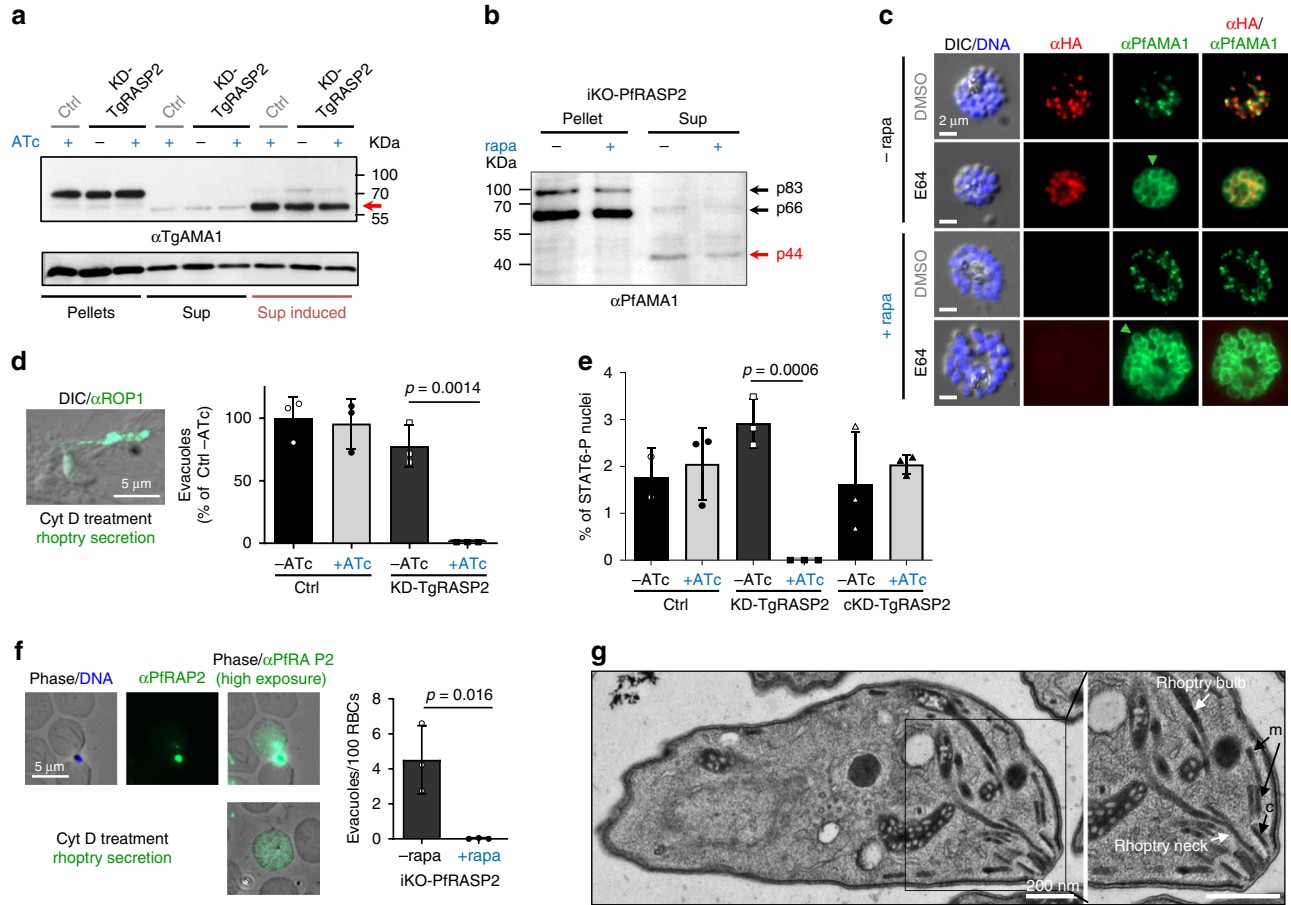

**Fig. 6** RASP2 plays an essential role in rhoptry secretion. **a** Immunoblot showing microneme secretion (arrow = processed/secreted TgAMA1) in TATi_TgRASP2-HA₃ (Ctrl) and KD-TgRASP2 parasites. Propranolol induced secretion (Sup induced). TgGRA3, loading control. **b** Immunoblot of iKO-PfRASP2-HA₃ culture supernatant (secreted proteins) showing microneme secretion (PfAMA1 p44) in control and PfRASP2-depleted parasites. Late schizonts were allowed to egress for 45 min in RPMI. Free merozoites and culture supernatant were separated by centrifugation and processed for Western blot and probed with rabbit anti-PfAMA1. **c** IFA on E64 or DMSO-treated iKO-PfRASP2-HA₃ parasites ± rapamycin. PfRASP2 depletion does not affect PfAMA1 relocalisation at the surface of merozoites (green arrows). **d** Left: IFA showing a representative example of an evacuole/rhoptry secretion event in *T. gondii* (ROP1 staining), Right: Quantification of evacuoles in TATi_TgRASP2-HA₃ (Ctrl) and KD-TgRASP2 parasites (±ATc) (One representative experiment out of three independent assays). **e** STAT6-P assay on the same lines and the complemented cKD-RASP2 line. **f** Left: IFA showing two representative examples of a rhoptry secretion event in *P. falciparum* (RAP2 staining). Right: Quantification of secretion events in iKO-PfRASP2 parasites (±rapamycin). Means ± SD for three independent experiments where >2,500 RBCs were analysed. **g** Ultrastructure of KD-TgRASP2 + ATc tachyzoites shows no defect in rhoptry positioning. m: micronemes; c: conoid. Source data are provided as a Source Data file

being washed in warm RPMI 1640 medium and returned to culture in complete medium.

**Microneme secretion assay**. Freshly egressed *T. gondii* tachyzoites (TATi_T-gRASP2-HA₃ and KD-RASP2-HA₃ pre-treated for 48 h ± ATc) were harvested by centrifugation at 600 g, RT for 10 min and washed twice in intracellular buffer (5 mM NaCl, 142 mM KCl, 1 mM MgCl₂, 2 mM EGTA, 5.6 mM glucose and 25 mM HEPES, pH 7.2) prewarmed to 37 °C. Parasites were resuspended in DMEM (supplemented with 2 mM glutamine) ± propranolol 500 μM and incubated 20 min at 37 °C. Parasites were centrifuged at 1000 × g for 5 min, 4 °C. Pellets were washed once in PBS and stored at −20 °C. Supernatants were centrifuged at 2000 × g for 5 min, 4 °C and supernatant was used as ESA (excreted/secreted antigen). Pellets and ESA samples were analysed for microneme (AMA1) and dense granule (GRA3) by Western blot.

**Rhoptry secretion assay**. To quantify rhoptry secretion in *Toxoplasma*, freshly egressed parasites (KD_RASP2-HA₃ pre-treated for 48 h ± ATc) were preincubated in 1 μM of cytochalasin D (cytD) for 10 min and then incubated for 15 min with HFF cells in the presence of cytD. IFAs were then performed with anti-ROP1 (rhoptry secretion) and anti-SAG1 (parasite surface), and the number of ROP1 stainings per field was determined by microscopic examination of at least 30 fields per coverslip (*n* = 3). For the method using STAT6-P detection in the host nucleus, see [36].

To quantify rhoptry secretion in *P. falciparum*, iKO-PfRASP2 (± rapa) purified schizonts were arrested with 1 μM Compound 2 for 2–4 h before being washed twice in order to allow them to egress. Parasites were then incubated for 30 min in complete medium with 1 μM Cyt D in the presence of RBCs. Samples were then processed for IFA using anti-PfRAP2 to visualise rhoptry secretion events (RAP2 export into the RBC). The number of secretion events were counted by microscopic examination of at least 2500 RBCs.

**Lipid dot blot**. Membrane lipid strips (Echelon P-6002) were incubated according to the manufacturer's recommendations for 2 h with either recombinant TgRASP2con2-His (0.5 μg/ml) or a His-tagged negative control protein (PfA-MA1D₁D₂D₃[37]) (at the same molarity). Strips were then incubated with mouse anti-His 1:3000 for 2 h and anti-mouse HRP 1:10,000 for 1 h. Detection was performed with TMBP precipitating agent (Echelon K-TMBP).

**Liposome binding assays**. *Preparation of liposomes*: Lipids DOPA, DOPE, DOPC, Brain PI(4,5)P and Brain PI(4)P were obtained commercially (Avanti) as chloro-form dissolved lipids. Volumes of lipids were separately pipetted into glass vials. Chloroform was removed by evaporation under a nitrogen stream and by desic-cation overnight. Lipids were re-suspended in lipid buffer 1 (25 mM Hepes pH 7.5, 100 mM NaCl, 5% glycerol, filtered 0.22 μm) and vortexed at room temperature for 5 min to generate a cloudy solution. The solution was bath-sonicated for 5 min and freeze-thawed 5 times. Lipids were extruded by passing them 11 times through a 0.1 μm filter.

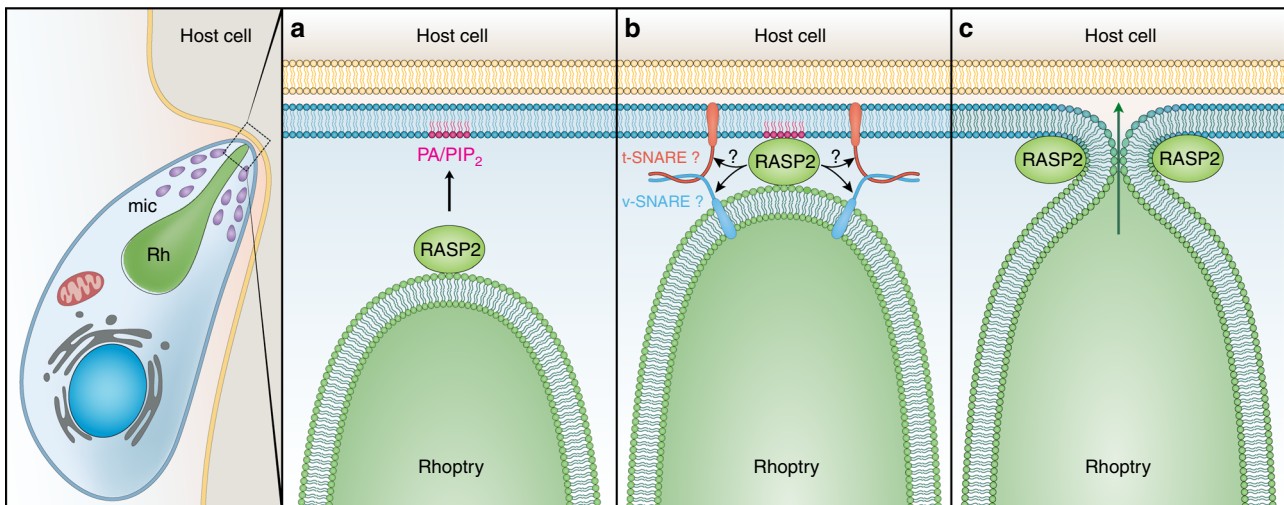

**Fig. 7** RASP2 contains lipid binding domains necessary for rhoptry secretion. **a** Schematic representation of TgRASP2 domains and the TgRASP2con2 recombinant expression construct. **b** Immunoblots of recombinant TgRASP2con2 and TgRASP2con2$^{MUT}$ associated or not with liposomes containing phosphatidylethanolamine (PE) and phosphatidylcholine (PC) (proportions 10%: 90%) phosphatidic acid (PA), PE and PC (proportions 30%: 10%: 60%) and phosphatidylinositol-4,5-bisphosphate (PI(4,5)P$_2$), PE and PC (proportions 25%: 10%: 65%). Ctrl, no lipid control; SN, supernatant fraction (unbound); P, pellet fraction (bound protein); Immunoblots were probed with mouse anti-His antibodies. Blots are representative of ≥ three independent experiments. **c** Quantification of liposome binding assays. p values generated from Student's t test. Error bars represent ±SD from three or more replicates. **d** Rhoptry secretion assay with Cpt_WT, ΔPH, L3* and the double mutant ΔPH_L3*. One representative experiment out of three independent assays. Source data are provided as a Source Data file

**Fig. 8** Model of RASP2 rhoptry docking/priming function in apicomplexan parasites. Upon response to an unknown signal, RASP2 binds to phospholipids (PA/PIP2) (**a**). This promotes association of the rhoptry with the parasite plasma membrane to initiate the assembly of the docking/priming machinery of the rhoptry (**b**), which results in membrane fusion, an essential step before the release of the rhoptry contents into the host cell (**c**)

Liposomes of the following composition were prepared (weight %)
10%: 90% - DOPE: DOPC
30%: 10%: 60% - DOPA: DOPE: DOPC
30%: 10%: 60% - PI(4)P: DOPE: DOPC
25%: 10%: 65% - PI(4,5)P$_2$: DOPE: DOPC

*Binding assays*: Recombinant TgRASP2con2 and TgRASP2con2$^{MUT}$ were centrifuged for 30 min at 100,000 × g at 4 °C to remove any potential precipitation. The soluble protein was transferred to a fresh tube and diluted to 40 μg/ml in protein buffer (20 mM HEPES, pH 7.5, 150 mM NaCl, 1 mM DTT). Lipids where diluted to 1 mg/ml using lipid buffer 2 (50 mM Hepes pH 7.4, 100 mM NaCl, 1% glycerol) and diluted 2-folds in water. Lipid binding reaction was initiated by mixing 40 μl of lipids (=40 μg) with 40 μl of protein (=1.6 μg) on ice for 1 h. The mixture was then centrifuged at 100,000 × g for 30 min at 4 °C and the supernatant and pellet were separated and resuspended in Laemmli buffer before being processed for SDS–PAGE. Blots were developed with anti-His antibody and bands were analysed with Image Lab software (BioRad).

**Organelle topology/Proteinase K protection assay**. The protocol used was derived from Bullen et al[27]. Freshly egressed *T. gondii* tachyzoites (TgRASP2-HA$_3$ or TgRASP1-HA$_3$) were resuspended in 4.5 ml cold SoTE (0.6 M sorbitol, 20 mM Tris–HCl [pH 7.5], and 2 mM EDTA) and split into 9 tubes (0.5 ml each). Cold SoTE was added to tube 1 as a control. Tubes 2 to 5 and 2′ to 5′ were permeabilized with 0.5 ml cold SoTE with 5%, 2.5%, 1% and 0.5% digitonin (Fluka 37006). Samples were carefully mixed by inversion and incubated on ice for 5 min prior to centrifugation (600 × g/4 °C/10 min). Supernatant was discarded, and 0.5 ml cold SoTE was added to tube 1, while 2.5 μl of cold Proteinase K (Sigma, 20 mg/ml)/SoTE was added to tubes 2′-5′. All tubes were gently inverted and incubated on ice for 30 min. Proteinase K was inactivated by addition of cold trichloroacetic acid to a final concentration of 10% on ice for 30 min. Samples were centrifuged (14,000 rpm/20 min), washed 2× with acetone, air dried, and resuspended in TE and Laemmli buffer prior to SDS–PAGE.

**Electron microscopy**. KD-TgRASP2-HA$_3$ parasites, and the Δku80-TATi_T-gRAPS2-HA$_3$ parental strain were treated for 3 days with ATc. Extracellular parasites were collected in the culture medium after natural egress and fixed by adding an equal volume of phosphate buffer 0.1 M containing 5% of glutaraldehyde for one hour at RT. Parasites were then centrifuged and resuspended in 1 ml of fresh buffer with 2.5% glutaraldehyde for 2 h before being kept at 4 °C until further processing. All the following steps were performed in suspension followed by centrifugation steps in a tabletop microcentrifuge. Sample were post-fixed with 1% OsO$_4$ and 1.5% potassium ferricyanid in 0.1 M phosphate buffer for 1 h at RT, washed in water and then incubated in 2% uranyl acetate in water overnight at 4 °C. Tachyzoites were then dehydrated in growing concentration of acetonitrile, followed by impregnation in Epon118: acetonitrile 50:50 for 2 h, 90:10 2 h and then overnight in pure Epon. Pellets were then allowed to polymerize in fresh Epon 48 h at 60 °C. 70 nm ultrathin sections were cut with a Leica ultracut (Leica microsystems), counterstained with uranyl acetate and lead citrate.

For immune-electron microscopy of TgRASP2-HA$_3$ parasites, infected fibroblast monolayers were trypsinized and fixed by adding an equal volume of 8% formaldehyde (FA) in phosphate buffer overnight at 4 °C and resuspended in 4 % of fresh FA until further processing. Cells were then incubated in 0.1% glycine in phosphate buffer, pelleted and embedded in 12% gelatine, cut in small blocks (< 1 mm) and infused 24 h in 2.3 M sucrose on a rotating wheel at 4 °C. Gelatine blocks were mounted on specimen pins and frozen in liquid nitrogen. Cryo-sectioning was performed on a Leica UC7 cryo-ultramicrotome, 70 nm cryosections were picked-up in a 1:1 mixture of 2.3 M sucrose and 2% methylcellulose in water and stored at 4 °C. For on-grid immunodetection, grids were floated on PBS 2% gelatine 30 min at 37 °C to remove methylcellulose/sucrose mixture, then blocked with 1% skin-fish gelatine (SFG, Sigma) in PBS for 5 min. Successive incubation steps were performed on drops as follows: 1) rat monoclonal anti-HA (clone 3F10, Roche) in 1% BSA, 2) rabbit polyclonal anti-rat IgG antibody (Sigma) in PBS 1 % BSA, 3) Protein A-gold (UMC Utrecht) in PBS 1% BSA. Four 2 min washes in PBS 0.1% BSA were performed between steps. After Protein A, grids were washed 4 times 2 min. with PBS, fixed 5 min in 1% glutaraldehyde in water then washed 6 times for 2 min with distilled water. Grids were then incubated with 2% methylcellulose: 4% uranyl acetate 9:1 15 min on ice in the dark, picked-up on a wire loop and air-dried.

All chemicals were from Electron Microscopy Sciences (USA), solvents were from Sigma. Observations and image acquisition were performed on a Jeol 1200 EXII transmission electron microscope on the Electron Microscopy Platform of the University of Montpellier (MEA; http://mea.edu.umontpellier.fr). Transmission electron microscopy images were processed with Fiji for contrast optimization and the Figure J plugin was used to make the EM panels.

**Super-resolution microscopy**. HFF host cells were inoculated onto sterile round coverslips and infected with a drop of TgRASP1-HA$_3$ tachyzoites. After 24–36 h, coverslips were fixed with 3.2% FA in PBS, permeabilized with 0.2% Triton X-100, washed three times then blocked with 3% BSA in PBS for 30 min. Coverslips were then incubated with primary antibody (HA-rat 3F10 1:300, RON9-mouse 1:400,

IMC1-rabbit 1:5000) in 3% BSA in PBS overnight, washed three times with PBS then incubated with secondary antibodies (Alexa Fluor 546, 488, 405 at 1:600) in 3% BSA/PBS for one hour. After secondary incubation, coverslips were washed 3 times and mounted onto glass slides with Prolong Diamond. Super-resolution microscopy was performed with a Zeiss ELYRA S1 (SR-SIM) microscope with a high-resolution Axio observer Z1 inverted microscope stand with transmitted (HAL), UV (HBO) and high-power solid-state laser illumination sources (405/488/561 nm), a ×100 oil immersion objective, and an Andor iXon EM-CCD camera. Images were acquired with ZEN software (Zeiss) with a SIM analysis module and analysed with ImageJ.

**Statistical analysis**. All results are presented as mean values with standard deviations shown as error bars. No statistical tests were used to predetermine sample size. Two-tailed Student's *t* tests was used appropriately to determine statistical significance for invasion, replication, rhoptry secretion, and egress assays. A *p*-value of 0.05 was considered significant.

**Protein production, and purification**. The constructs encompassing both C2 and PH domains of TgRASP2 (TgRASP2con2: Gly45-Glu338) or the equivalent mutated version (TgRASP2con2$^{MUT}$: Lys61, Arg73, Lys123, Lys128, Lys129, Lys270, Lys273, Lys275, Arg291, Lys292, Arg294, Lys304 mutated to Asp) were codon optimized for expression in insect cells and synthesized by Genscript. The sequences were subcloned into a modified pAc vector (Pharmingen) incorporating an N-terminal hexahistidine (His$_6$) tag separated from the protein of interest by a TEV cleavage site. For large-scale protein production, *Trichoplusia ni* insect cells were infected with amplified virus. Following a 48 h infection, the cells were harvested by centrifugation and the pellets were resuspended in 20 mM HEPES pH 8.0, 1 M NaCl, 30 mM imidazole supplemented with 1 mM TCEP. Cells in suspension were lysed using French press, insoluble material was removed by centrifugation and the soluble fraction was allowed to batch-bind with nickel-Sepharose beads at 4 °C for 1 h. The proteins were eluted with buffer containing 250 mm imidazole and 1 mM TCEP, and fractions were analysed by SDS-PAGE and pooled based on purity. The His$_6$ tag was removed by TEV cleavage and the protein was further purified by size exclusion chromatography (HiLoad 16/60 Superdex 75; GE Healthcare) in HEPES-buffered saline (20 mM HEPES, pH 7.5, 150 mM NaCl) supplemented with 1 mM TCEP.

**Reagents and antibodies**. *Reagents*: Cytochalasin D (Sigma C8273) was prepared at 1 mM in DMSO and used at 1 μM final concentration. ATc (Fluka 37919) was prepared in 1 mg/ml in ethanol and used at 1 μg/ml final. Indolacetic acid (IAA) (ChemCruz sc-215171) stock was prepared at 1 M in methanol and used at 1 mM final concentration. Propranolol (Sigma P0884) was prepared at 100 mM in DMSO and used at 500 μM final concentration. A23187 stocks prepared at 2 mM and used at 3 μM final concentration. Compound 2 (4-[7-[(dimethylamino)methyl]-2-(4-fluorphenyl)imidazo[1,2-α]pyridine-3-yl]pyrimidin-2-amine (C2) was a kind gift from Dr. O. Billker (Sanger Institute, Cambridge, UK), stock solution was prepared in DMSO (3 mM) stored at −20 °C, and was used at final concentrations of 1 μM. Rapamycin (rapa) and E64 were obtained from Sigma (catalog numbers R0395 and E3132); stock solutions (100 μM for rapa and 50 mM E64) were prepared in DMSO and stored at −20 °C and were used at final concentrations of 10 nM (rapa) and 50 μM (E64). The antifolate drug WR99210 was from Jacobus Pharmaceuticals (New Jersey, USA) and used at 2.5 nM final concentration. Percoll was purchased from GE Healthcare (catalog number 17-0891-01) and diluted in 1/10 (v/v) of 10x PBS to obtain isotonic Percoll. Lipids were purchased as liquid chloroform/methanol solutions from Avanti: DOPC (850375C), DOPE (850725C), DOPA (840875C), Brain PI(4,5)P2 (840046X), Brain PI4P (840045X).

*Antibodies*: The antibodies used and their dilution for Western blot (WB) and immunofluorescence (IFA) were as follows: mouse mAb T4 1E5 anti-*Toxoplasma* SAG1, 1:2000 (IFA and WB)[38], mouse mAb T5 2A7 anti-*Toxoplasma* RON9, 1/200 (IFA)[39], rabbit anti-*Toxoplasma* ROP1, 1:3000 (IFA)[36], rabbit anti-*Toxoplasma* ARO, 1:500 (IFA)[17], mouse mAb T5 3E2 anti-*Toxoplasma* ROP5, 1:200 (IFA)[40], rabbit anti-*Toxoplasma* AMA1, 1:5000 (WB)[41], rabbit anti-*Toxoplasma* GAP45, 1:9000 (IFA)[42], mouse anti-*Toxoplasma* GRA3 T6 2H11 1D1 hybridoma, 1: 100 (IFA)[43], mouse mAb 24C6 4F12 anti-*Plasmodium falciparum* RON4, 1:200 (IFA)[44], mouse anti-*Plasmodium* MSP1, 1:1000 (IFA)[45], mouse mAb 63.1 anti-*Plasmodium falciparum* RhopH2, 1:200 (IFA)[46], rabbit anti-*Plasmodium falciparum* Aldolase (Abcam, ab207494), 1:1000 (WB), rabbit anti-*Plasmodium falciparum* AMA1, 1:1000 (IFA and WB)[47], mouse anti-*Plasmodium falciparum* AMA1 F8.12.19, 1:500 (WB)[48], mouse anti-*Plasmodium falciparum* RAP2, 1:500 (IFA)[49], mouse mAb anti-Ty, 1:200 (WB), 1:100 (IFA)[50], mouse hybridoma supernatant anti-c Myc 9E10, 1:10 (IFA and WB)[51], rat anti-HA 3F10 (Roche), 1:500–1:1000 (IFA and WB), mouse anti-His (Sigma), 1:3000 (WB), rabbit anti-Phospho-Stat6 (Tyr641) (CST, 56554 S), 1:600 (IFA).

For IFA studies, the secondary antibodies used were Alexa Fluor 488 and 594-conjugated antibodies against mouse, rat or rabbit IgG (highly cross-adsorbed) diluted as recommended by the manufacturer (Molecular Probes). For immunoblots, the secondary rat, mouse or rabbit antibodies used were coupled to alkaline phosphatase (Promega) or Horseradish Peroxidase (Jackson IR).

**Immunoblots and Immunofluorescence (IFA) microscopy**. All gels were run under reducing conditions. Wet transfer was performed for Western blotting and immunodetection following standard protocols.

For IFAs of intracellular tachyzoites grown in HFFs, cells were fixed with 4% formaldehyde (FA) in Phosphate Saline Buffer (PBS), quenched with 0.1 M Glycine in PBS and permeabilized with 0.1% (v/v) Triton X-100 in PBS. Coverslips were blocked in PBS 10% FCS or 1.5 % BSA and proceeded further for IFA by incubating with relevant primary antibodies diluted in blocking solution and with appropriate secondary antibodies before counterstaining with Hoechst. For invading parasites, samples were permeabilized with saponin 0.1%. For IFA on *P. falciparum*, formaldehyde-fixed thin blood smears were permeabilized with 0.1 % (v/v) Triton X-100 and blocked in 1.5% BSA. Samples were then processed as described above for *T. gondii*. Observations were performed with a Zeiss Axioimager epifluorescence microscope equipped with an apotome and a Zeiss Axiocam MRmCCD camera. *P. falciparum* merozoites were captured with a Zeiss LSM 880 equipped with an Airyscan detector to improve the optical resolution of the scanned images at the Montpellier RIO imaging facility. Super-Resolution images were acquired using the Zeiss ELYRA S1 (SR-SIM) microscope. Fluorescence microscopy and super-resolution images were processed using Zen Blue 2.3 pro (Zeiss). The frontal orthogonal projection shown in Fig. 1b was obtained using the weighted average orthogonal projection method, and the 3D surface reconstruction was made using Zen 3D Visualisation module.

**Cloning of DNA constructs**. Unless otherwise stated, all PCR amplifications were performed with the Phusion polymerase (NEB) and the primers are listed in the Supplementary Table 2.

Tagging vectors and strategies: The plasmids pLIC_DHFR_TgRASP1-HA₃ and pLIC_DHFR_TgRASP2-HA₃ were designed to express TgRASP1 and TgRASP2 with a triple hemagglutinin (HA) tag in Δku80 parasites. The 3' end of the genomic sequences of *tgrasp1* and *tgrasp2* were PCR amplified with primers 1123/1124 and 1418/1419, respectively, cloned in-frame with a triple HA tag in the pLIC-DHFR-HA₃[33] and linearized by XhoI (pLIC_DHFR_TgRASP1-HA₃) and BclII (pLIC_DHFR_TgRASP2- HA₃) prior to transfection. Single homologous recombination (knock in) at the endogenous locus allowed endogenous tagging of *tgrasp1* and *tgrasp2*. The resulting parasite lines were designated as TgRASP1-HA₃ and TgRASP2-HA₃.

To tag TgRASP1 with a triple c-Myc tag in the KD-RASP2 line, the template vector pCR2.1_TgRASP1recodMyc₃-3'UTR was amplified with the KOD polymerase (Novagen) and used as a donor sequence. For this, primers 2739/2742 were used to amplify a sequence which contained 227 bp of TgRASP1 3' sequence (HR1) followed by 82 bp of TgRASP1 recodonised sequence, a triple c-Myc tag and 648 bp of 3'UTR (HR2). A guide RNA cutting 66 bp upstream of the *tgrasp1* stop codon was generated by cloning the annealed primers 2066/2067 into the BsaI-digested pU6-Cas9-YFP and named pU6-Cas9-YFP-TgRASP1b. The PCR product corresponding to the donor fragment was cotransfected with the pU6-Cas9-YFP-TgRASP1b allowing integration of the triple c-Myc tag into the TgRASP1 locus. The resulting parasite line was designated as KD-TgRASP2/TgRASP1-myc₃.

To Tag TgRASP2 with a triple HA tag in the KO-TgRASP1 line, the 3' end of genomic sequences of *tgrasp2* was PCR amplified with primers 1418/1419 respectively and cloned in-frame with a triple HA tag in the pLIC_CAT-HA₃ vector[33]. Before transfection the resulting plasmid pLIC_CAT_TgRASP2-HA₃ was linearized at a unique PstI restriction site that lies in the middle of the *tgrasp2* sequence in order to integrate the vector by single homologous recombination. The resulting parasite line was designated as KO-TgRASP1/TgRASP2-HA₃.

To tag TgRASP3 with a triple Ty tag in the TgRASP2-HA₃ line, the template vector pTgCtermMLC2g_3T[52] containing a triple Ty tag followed by a generic 3' UTR and a HXGPRT resistance cassette was amplified with the KOD polymerase (Novagen). For this, primers 2946 and 2947 containing 30 bp homology to the 3' end of TgRASP3 were used. The pU6-Cas9-TgRASP3tag plasmid, which targets the 3'UTR of the *tgrasp3* gene 50 bp downstream of the stop codon, was constructed by annealing primers 2972 and 2973 and ligating them in the BsaI-digested pU6-Universal vector (gift from S. Lourido). The PCR product corresponding to the donor fragment and the pU6-Cas9-TgRASP3tag were co-transfected in the Δku80 TgRASP2-HA₃ line allowing integration of a triple Ty tag into the TgRASP3 locus. The resulting parasite line was designated as TgRASP2-HA₃/TgRASP3-Ty₃.

To tag PfRASP2, a 5' UTR homology region (HR1) of 1244 bp was amplified with primers 2748/2749 from gDNA as well as 280 bp of recodonised PfRASP2 3' sequence with primers 2750/2751. These two PCR products were cloned by In-Fusion HD (Clontech 638918) into the pLN_BSD_glmS_PbDT3' plasmid previously digested with StuI/AfeI. The recodonised sequence of PfRASP2 was ordered as a synthetic gBlock fragment (IDT DNA) and used as a template. In a second step, a 3' UTR homology region (HR2) of 518 bp was amplified with primers 2752/2753 and cloned by In-Fusion HD into the pLN_BSD_TgRASP2-HA₃_glmS_PbDT3'_HR1 previously linearized with PstI. pLN_BSD_HA3-glmS_PbDT3' was generated by cloning a gBlock containing a triple-HA tag and the riboswitch glmS sequence into the pLN vector[53] digested BamHI/HpaI. The PbDT3' was removed from the pGFP-glmS[54] and cloned into the pLN-BSD-glmS digested with XhoI/NotI. The resulting parasite line was designated as PfRASP2-HA₃.

Knock down and knock out vectors: To knock out the *tgrasp1* gene in the Δku80 TgRASP1-HA₃ line, the donor plasmid pKO-TgRASP1 was generated by cloning 900 bp and 1 kb of homology regions (HR) on either end of the HXGPRT selection marker in the pminiHXGPRT vector[55]. PCR amplifications were performed with primers 1585/1586 (HR1) and 1587/1588 (HR2). First, HR1 was ligated into HindIII/XhoI digested pminiHXGPRT. Second, HR2 was ligated in the BamHI/XbaI-digested pminiHXGPRT_HR1. A guide RNA cutting 66 bp upstream of the *tgrasp1* stop codon was generated by cloning the annealed primers 2066/2067 into the BsaI-digested pU6-Universal. The resulting plasmid (pU6-Cas9-RASP1KO) was co-transfected with the vector pKO-TgRASP1 previously linearized with XhoI and XbaI. The resulting parasite line was designated as KO-TgRASP1.

To generate a conditional knock-down of TgRASP3, an auxin-induced degradation strategy was designed. A template vector containing the DHFR resistance cassette and the auxin-inducible degron (AID) fused to triple HA tag was PCR amplified (KOD) with primers 3423 and 3428 flanked by 30 bp homology arms to the *tgrasp3* 3' UTR. The PCR product corresponding to the donor fragment and the pU6-Cas9-TgRASP3tag plasmid (which target the 3'UTR of *tgrasp3*) were co-transfected in Δku80 parasites expressing the auxin receptor TIR1. The resulting parasite was designated as KD-TgRASP3.

To generate a conditional knock-down of TgRASP2 in Δku80-TATi parasites, a Tet-OFF system strategy was designed based on plasmid DHFR-TetO7-SAG4 used in the Δku80-TATi1 strain[34]. We first introduced a C-terminal HA tag to TgRASP2 in the Δku80-TATi line using plasmid pLIC_CAT_TgRASP2-HA₃. The resulting parasite was designated as TATi-TgRASP2-HA₃. A guide RNA, pU6-Cas9-TgRASP2prom, cutting 146 bp upstream of the *tgrasp2* start codon was generated by cloning the annealed primers 2418/2419 into the BsaI-digested vector pU6-Universal. The vector DHFR-TetO7-SAG4 containing a DHFR resistance cassette and the ATc repressible TetO7-SAG4 promoter was used as a template. A PCR (KOD) was performed with primers 2420/2421 both flanked by 30 bp homology arms to the 5'UTR (2420) and the beginning of the *tgrasp2* CDS (2421) were used. The PCR product corresponding to the donor fragment and the pU6-Cas9-TgRASP2prom plasmid were co-transfected in the TATi-TgRASP2-HA₃ line allowing expression and regulation of *tgrasp2* under the TetO7-SAG4 promoter. The resulting parasite was designated KD-TgRASP2.

To construct a conditional deletion vector for PfRASP2. The plasmid pLN_BSD_TgRASP2-HA₃glmS_PbDT3'_HR2, was digested with XmaI and XhoI to remove the glmS sequence but conserve the generic 3'UTR (PbDT3'). Three PCR products corresponding to 699 bp of the *pfrasp* promoter (primers 3179/3180), a second PCR of 246 bp (primers 3181/3182) corresponding to the exon1 and LoxPint[56] and a third PCR corresponding to the recodonised ex2-5 and HA₃ (primers 3183/3184 on gBlock mentioned above, ordered to tag *pfrasp2*) were assembled by In-Fusion HD into the final vector named pFloxed_PfRASP2-HA₃. The first exon of *pfrasp2* and a LoxPint replacing the first endogenous intron was ordered as a synthetic gBlock (IDT DNA) and cloned into a TOPO-Blunt vector (Invitrogen). The assembly of the three PCRs resulted in a floxed recodonised *pfrasp2* cDNA sequence in addition to a triple HA tag. The resulting parasite line was designated as iKO-PfRASP2.

Complementation vectors: To complement the KD-RASP2 line with a wild type copy of *tgrasp2* under its own promoter, the pUPRT-promRASP2-TgRASP2cdna-Myc was generated and integrated at the UPRT locus by double homologous recombination and using the FUDR negative selection[34]. For this, 1749 bp of the *tgrasp2* promoter was amplified with primers 2682/2683, as well as the complete cDNA sequence of TgRASP2 with primers 2684/2688. Both these PCR products contained between 15 and 18 nucleotides of homology in order to be assembled by In-Fusion HD cloning into pUPRT-RON5-G13-Myc vector[57] digested with NotI and EcoRV. A guide RNA cutting 107 bp downstream of the *tguprt* stop codon was generated by cloning the annealed 2087/2088 primers into the pU6-Cas9-Universal. The resulting CRISPR/Cas9 vector was co-transfected with the pUPRT-TgRASP2prom-TgRASP2cdnaMyc linearized with NsiI and KpnI. The resulting parasite was designated as cKD-TgRASP2.

To complement the KD-TgRASP2 line with a wild type copy of *tgrasp2* under the tubulin promoter, the complete cDNA sequence of *tgrasp2* and a triple Ty tag was amplified with primers 3542/3543. This PCR product contained between 15 and 18 nucleotides of homology in order to be assembled by In-Fusion HD cloning into pUPRT-promTUB-G13-Ty vector[34] digested with BglII and EcoRV. The resulting CRISPR/Cas9 vector was co-transfected with a PCR product generated from the pUPRT-promTUB-TgRASP2cdnaTy₃ with primers 2865/2866. The resulting parasite line was designated as cKD-pTUB_TgRASP2.

For the complementation containing the three potentially palmitoylated cysteines mutated to alanines, a similar strategy to that described above was used. The 5' cDNA sequence of TgRASP2 containing the three point mutations was ordered as a synthetic gBlock fragment (IDT DNA) and used as template. This fragment was amplified with primers 3542/3544 giving a 500 bp product that was assembled in the same vector as described above by In-Fusion HD cloning with the 3' end of TgRASP2 cDNA (1044 bp) amplified with primers 3545/3543. The resulting parasite line was designated as KD-RASP2_cptRASP2_C₂₃C₂₃C₃₀/A-Ty₃.

For the complementation containing modifications in the predicted Loop3 ¹²¹EVKIFLFKK¹²⁹ of the putative C2 domain of TgRASP2, a similar strategy to that described above was used. The synthetic gBlock contained triple point mutations of the lysines to aspartic acid residues at positions K123/K128/K129 to generate pUPRT-promTUB-TgRASP2_L3^K/D_Ty₃. This gBlock was

amplified with primers 3542/3544 giving a 500 bp product that was assembled in the same vector as described above by *In-Fusion HD* cloning with the 3′ end of TgRASP2cDNA (1044 bp) amplified with primers 3545/3543. The resulting parasite line was designated as L3*.

For the complementations that lack the putative C2 and PH domains as well as the stretch of polybasic residues (PBR), two PCR products were assembled by *In-Fusion HD* in the pUPRT-promTUB-G13-Ty vector[34] digested with BglII and EcoRV. For the vector with TgRASP2 lacking the putative C2 domain, PCRs were performed with primers 3542/3396 and 3397/3543. For the vector lacking the putative PH domain, PCRs were performed with primers 3542/3399 and 3400/3543. For the vector lacking the PBR stretch, PCRs were performed with primers 3542/3401 and 3402/3543. The resulting vectors where used as template to generate the double and then triple mutants lacking multiple domains. The resulting parasite lines were designated as ΔC2, ΔPH, ΔPBR, ΔPH_ΔPBR, ΔPH_L3*, ΔPBR_L3* and ΔPH_ΔPBR_L3*.

**Transfection and selection of transformants**. For *Toxoplasma*, $14 \times 10^6$ *T. gondii* tachyzoïtes were transfected by electroporation at 2,02 kV, 50Ω, 25μF in an Electro Cell Manipulator 630 (BTX) with 30 μg donor plasmid DNA or 5 μg of PCR products obtained with the KOD polymerase in addition to 30 μg of CRISPR/Cas9 plasmid. All donor plasmids used were linearized prior transfection. Transgenic parasites were selected by addition of pyrimethamine at 1 μM for DHFR integration; mycophenolic acid at 25 μg/ml plus xanthine at 50 μg/ml for HXGPRT integration; FUDR at 5 μM for UPRT loss and 5 μg/ml of phleomycin for bleomycin selection. For marker-free transfections (tagging TgRASP1 in KD-RASP2 line and tagging TgRASP3 in TgRASP2-HA3 line), parasites transiently expressing Cas9-YFP were sorted by Flow cytometry (FACS) two days after transfection and directly cloned into a 96-well plate. For each transfection, clones were screened by PCR for correct vector integration.

For generation of marker-free, DiCre transgenic *P. falciparum* parasites 5–10 % ring stages were transfected with 60 μg of linearized rescue plasmid and 25 μg of CRISPR/Cas9 circular plasmid. Transgenic parasites were grown in agitation (200 rpm) and selected by addition of 2.5 nM WR99210 for 7 days. When genome integration was detected by diagnostic PCR, parasites were cloned by limiting dilution.

**Invasion, Replication and plaque assays**. Invasion, Replication and plaque assays of *T. gondii* tachyzoites were performed as standard assays[36].

*P. falciparum* experiments have been performed on tightly synchronized parasites with a 1 h re-invasion window. To follow *P. falciparum* intra-erythrocytic development, synchronized parasite cultures were smeared 1 h post-invasion until 54 h (second cycle/reinvasion). Proliferation rate/invasion assays were set up at 0.5–1% parasitaemia in ring stage[58]. Samples were taken up in ring stage 4–6 h post-invasion during the first cycle. 44 h later, ring stage samples of the next cycle were again collected and fixed in 4% FA for 4 h at room temperature. FACS was then used to determine parasitaemia. Fixed cells were washed twice in PBS, followed by 30 min (min) incubation with 1 X SybrGreen (Invitrogen) in the dark. Cells were washed, resuspended in 700 μl PBS and analysed by BD FACS Canto I flow cytometer using FACS Diva software (BD Biosciences). SYBR green was excited with a blue laser at 488 nm, and fluorescence was detected by a 530/30 nm filter. The invasion rate was calculated by dividing the number of events (rings) obtained in the second cycle by the ones obtained in the first cycle. Graph showing the invasion rates relative to the control - rapamycin.

**Motility assay and egress**. Motility and egress of *Toxoplasma* was quantified by standard methods[59]. *P. falciparum* egress was imaged using 1 μM C2 to tightly synchronize egress. Following removal of C2 (2 RPMI washes), parasites were suspended in warm medium and introduced into a 1μ-Slide I0.2 Luer chamber slide (Ibidi 80166) on a temperature-controlled microscope stage at 37 °C. DIC images were collected 3 min after washing off the C2 at 5 s intervals for 10–15 min using a ZEISS AxioObserver Microscope fitted with a coolsnap HQ2 digital camera. Images were converted to MOV movies using ZEN software.

**Yeast-two-hybrid analysis**. Yeast two-hybrid screening was performed by Hybrigenics Services, S.A.S., Paris, France (http://www.hybrigenics-services.com). The coding sequence of the full-length *T. gondii* RASP2 protein (TGME49-315160) was PCR-amplified and cloned into pB29 as a C-terminal fusion to LexA (TgRASP2-LexA) DNA-binding domain. The constructs were checked by sequencing and used as baits to screen a random-primed *T. gondii* wt RH strain cDNA library constructed into pP6. pB29 derives from the original pBTM116 vector[60]. For the TgRASP2-LexA bait construct, 57 million (5-fold the complexity of the library) of clones were screened using a mating approach with YHGX13 (Y187 *ade2-101::loxP-kanMX-loxP, MAT*α) and L40αGal4 (*MAT*α) yeast strains[61]. 27 His+ colonies were selected on a medium lacking tryptophan, leucine and histidine. The prey fragments of the positive clones were amplified by PCR and sequenced at their 5′ and 3′ junctions. The resulting sequences were used to identify the corresponding interacting proteins in the GenBank database (NCBI) using a fully automated procedure. A confidence score (PBS, for Predicted Biological Score) was attributed to each interaction[62]. The confidence scores rank from

A (highest confidence) to F (technical false-positive) and have been shown to positively correlate with the biological significance of interactions[63–65].

**Co-immunopurification**. Parasites were solubilized in lysis buffer (1% NP40, 50 mM Tris-HCl pH 8, 150 mM NaCl, 4 mM EDTA and protease inhibitor) and immunoadsorption procedures were performed using magnetic beads coupled to anti-HA antibodies (Roche 11815016001). After overnight incubation of the lysate with HA beads, these were washed 5 times in wash buffer (50 mM Tris pH 8, 150 mM NaCl and 0.5% NP40). Elution from beads was achieved by adding Laemmeli buffer and incubating 5 min at 95 °C. Western blots were performed on the eluates with anti-HA and anti-c-Myc antibodies.

**Bioinformatics predictions**. SignalP software (http://www.cbs.dtu.dk/services/SignalP/) was used to predict signal peptides. The C2 and PH domains of TgRASP2 were predicted by using Phyre2 (Protein Homology/analogy Recognition Engine) (http://www.sbg.bio.ic.ac.uk/~phyre2/). Alignment was performed using MUSCLE (http://www.drive5.com/muscle/muscle.html).

**Homology modelling of the TgRASP2 C2 and PH domains**. Homology searches of TgRASP2 were first performed using Phyre2[23], which predicted C2 (lipid-binding calcium) and PH (Pleckstrin homology domain). These structural templates were then leveraged to generate energy-minimized models of TgRASP2 C2 and PH domains using Modeler 9v18[66]. The homology models for TgRASP2_C2 (Gly45–Leu179) and TgRASP2_PH (Glu230-Glu338) (ToxoDB-TGME49_315160) were based on the crystal structures of *Rattus norvegicus* C2 [PDB id: 2CJS (13% identity) and 2CJT (15% identity)][67] and of *Homo sapiens* PH domain [PDB id: 1UNQ (15% identity)][68], respectively. Secondary structure prediction algorithms (Psipred[69], Jpred[70], and RaptorX[71]) were incorporated to overcome limited sequence identity and further augment model accuracy. The final models of TgRASP2_C2 and TgRASP2_PH were selected based on the low value of the normalized discrete optimized protein energy value (zDOPE).

**Differential scanning fluorimetry**. The rTgRASP2con2 $Ca^{2+}$ binding capacity was investigated by thermal denaturation in the presence of SYPRO orange (Invitrogen, CA) using a 7500 real-time PCR system (Applied Biosystems, CA). Measurements were performed in a 96-well plate in 20 μl containing TgRASP2con2 (200 μM) in HEPES buffer (20 mM HEPES (pH 7.5), 150 mM NaCl, 1 mM DTT) in the presence of 1 mM $CaCl_2$ or EDTA (0–15 mM). Different concentrations (0–15 mM) for both $Ca^{2+}$ and EDTA were tested but for the sake of clarity only the 2 mM treatment curves are presented. The fluorescence intensity was monitored by increasing the temperature in 1 °C increments from 25 to 99 °C. The thermal melting point ($T_m$) was identified from the midpoint of each melting curve. All thermal shift reactions were performed in triplicate.

**Isothermal titration calorimetry**. Purified rTgRASP2con2 was dialyzed against 20 mm HEPES, pH 7.5, 150 mm NaCl, and 1 mm TCEP at 4 °C overnight. All ITC experiments were carried out at 25 °C on a MicroCal iTC200 instrument (GE Healthcare). The sample cell contained TgRASP2con2 (0.15 mM) and the syringe was loaded with $CaCl_2$ (1.5 mM). A total of 19 injections at 2 μl each were used. The data were processed using Origin software (MicroCal).

**Reporting summary**. Further information on research design is available in the Nature Research Reporting Summary linked to this article.

## Data availability
The data that support the findings of this study are available from the corresponding author upon reasonable request. The source data of gels and blots underlying Figs. 1d, 4b, 5b, d, 6a–b, 7b as well as Supp. Fig. 1c, Supp. Fig. 2b–c, Supp. Fig. 4a, Supp. Fig. 5b-c and h, Supp. Figure 6b and d, Supp. Figure 7b and e, Supp. Fig. 8b, Supp. fig. 9g, and Supp. fig. 10e are provided in Supplementary figs 11 and 12. The source data of graphs underlying Figs. 1a, 3b, 4e–f, h, 5e–g, 6d–f, 7c–d, Supp. Fig. 5d–f, Supp. fig. e–f, Supp. fig. 9e–f and Supp. fig. 10e are provided as a Source Data file.

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

## Acknowledgements

We are grateful to Michael White and Olivier Lucas (University of Florida, USA) for the bioinformatic screen based on *Toxoplasma* mRNA abundance over the cell cycle and for providing a list of proteins that have a cell cycle profile that matched with the periodic expression pattern of rhoptry proteins. We thank Nicolas Dos Santos Pacheco for his help in setting up the liposome binding assay. We thank Sebastian Lourido for the pU6-Universal plasmid, Artur Scherf for anti-PfRAP2 antibodies, Dominique Soldati-Favre for anti-TgARO antibodies, Michael Blackman for anti-PfAMA1, anti-PfSUB1, anti-RhopH2 and anti-MSP1 antibodies, Anthony Holder for anti-mTIP antibodies. We thank Oliver Billker for providing us with Compound 2. We are grateful to Olivier Silvie for critical reading of the manuscript. We thank the staff of the MRI-Cytometry at the Institute for Regenerative Medicine and Biotherapy for their assistance in sorting parasites. We acknowledge the imaging facility MRI, member of the national infrastructure France-BioImaging infrastructure supported by the French National Research Agency (ANR-10-INBS-04, «Investments for the future»). This research was supported by the Laboratoire d'Excellence (LabEx) (ParaFrap ANR-11-LABX-0024), by the Fondation pour la Recherche Médicale (Equipe FRM DEQ20130326508 and FRM EQ20170336725) to M.L.; by Canadian Institutes of Health Research grant 148596 to M.J.B. M.J.B. gratefully acknowledges the Canada Research Chair program for salary support. C.S. was supported by a Swiss National Science Foundation (SNSF) Early Postdoc.Mobility fellowship. We would like to thank Julien Marcetteau for the graphic design of Fig. 8.

## Author contributions

C.S. performed most of the work on *Toxoplasma* and generated all the *Plasmodium* data. G.L. performed the initial screen and tagging of TgRASP1 and TgRASP2, M.M. contributed to phenotypic analysis and lipid dot blot assays. E.A. tagged TgRASP3 and performed co-IPs of TgRASPs. L.B.-S. and M.C. performed the electron-microscopy and super-resolution respectively. R.R. produced the recombinant proteins and performed the calcium binding assays. A.C. and P.B. contributed to co-immunoprecipitation experiments. M.L. supervised the research. C.S. and M.L. wrote the paper with editorial support from M.J.B., G.L. and B.S. The data are included in the main manuscript and in the supplementary materials.

## Additional information

**Competing interests:** The authors declare no competing interests.

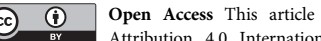

