## [Peer Review File · Nature Communications]

Editorial Note: This manuscript has been previously reviewed at another journal that is not operating a transparent peer review scheme. This document only contains reviewer comments and rebuttal letters for versions considered at Nature Communications .

Reviewers' Comments:

Reviewer #2:

Remarks to the Author:

The authors describe the discovery of RASP2 as a rhoptry protein required for rhoptry secretion and host cell invasion. Prior review comments focused on the need for more experimentation to support speculation on how RASP2 may mediate rhoptry secretion. Some additional studies have been provided.

The most recent version of the manuscript provides evidence for phospholipid binding by putative C2-PH domains of the protein. Fig 3f & g shows convincingly that the charge reversal mutations in the C2-PH domain abrogate binding to PA and PI(4,5)P2. Fig 3h shows the impact of mutation on rhoptry secretion although a different subset of mutations were used. In this case, C2 mutations combined with deletion of PH resulted in loss of secretion. One wishes that a common subset of mutations would have been used for both studies. Nonetheless the work supports the conclusions that RASP2 binds acidic phospholipids via its C2-PH domain, and that the C2-PH domain is required for rhoptry secretion.

It should be noted that the basis for assigning the C2 and PH domains remains indirect and not convincing.

A model is shown in Fig 4 where the rhoptry tip containing RASP2 is steered to the parasite plasma membrane by interactions with PA and PI(4,5)P2, which somehow sets up SNARE-dependent fusion of the rhoptry with the plasma membrane. In this model, "release of rhoptry contents into the host cell" should be changed to read "release of rhoptry contents onto the host cell" since the understanding of penetration of the host cell is still elusive. Are there any secreted rhoptry proteins that could form a fusion pore? See perforin.

Reviewer #3:

Remarks to the Author:

Since its last inception, the authors have shored-up their data now provide a strong argument that they have identified important machinery required for rhoptry discharge that also binds charged lipids. They show that proteins that are part of this complex have the most apical location of any known protein, likely located on the outer face of rhoptries, is required for invasion and release of rhoptries and, furthermore, binds PA and PI(3,4)P. The authors have toned down their language also to make sure that they do not overstate their conclusions. The authors (once again) should be commended for performing experiments in both *Toxoplasma* and *Plasmodium*.

This is the first evidence of the identity of the machinery that is required for rhoptry release, which has long been sought after. This work will trigger a range of new areas of investigation and will no doubt be highly cited.

UMR 5235 - D.I.M.N.P.

Dynamique des Interactions Membranaires Normales et Pathologiques

Our reply to the comments of reviewer 2 are found below:

- “A model is shown in Fig 4 where the rhoptry tip containing RASP2 is steered to the parasite plasma membrane by interactions with PA and PI(4,5)P2, which somehow sets up SNARE-dependent fusion of the rhoptry with the plasma membrane. In this model, “release of rhoptry contents into the host cell” should be changed to read “release of rhoptry contents onto the host cell” since the understanding of penetration of the host cell is still elusive.”

The rhoptry content is never secreted onto the host cell but injected in the host. We expanded the discussion to clarify that our study defines a role of RASP2 during the first exocytic step of the complex mechanism of rhoptry secretion (Fig. 8, panels A and B) but leaves unknown how the content of the organelle is injected into the host (panel C).

- “Are there any secreted rhoptry proteins that could form a fusion pore? See perforin.”

No secreted rhoptry proteins forming a fusion pore have been described in Apicomplexa. We also expanded the discussion to describe what is known on that subject to date.